# Optimizing Recombinant Baculovirus Vector Design for Protein Production in Insect Cells

Carina Bannach [1,*], Daniel Ruiz Buck [1], Genna Bobby [1], Leo P. Graves [1], Sainan Li [2,3], Adam C. Chambers [1], Elizabeth Gan [1], Raquel Arinto-Garcia [1], Robert D. Possee [1] and Linda A. King [2]

1 Oxford Expression Technologies Ltd., BioInnovation Hub, Gipsy Lane, Oxford OX3 0BP, UK; daniruizbuck@gmail.com (D.R.B.); g.bobby@oetltd.com (G.B.); l.graves@oetltd.com (L.P.G.); a.chambers@oetltd.com (A.C.C.); e.gan@oetltd.com (E.G.); raquelbag@gmail.com (R.A.-G.); r.possee@oetltd.com (R.D.P.)
2 Department of Biological and Medical Sciences, Oxford Brookes University, Oxford OX3 0BP, UK; lisainan2001@sina.com (S.L.); laking@brookes.ac.uk (L.A.K.)
3 Department of Biology, Zhaoqing University, Zhaoqing 526061, China
* Correspondence: c.bannach@oetltd.com

**Abstract:** Autographa californica nucleopolyhedrovirus is a very productive expression vector for recombinant proteins in insect cells. Most vectors are based on the polyhedrin gene promoter, which comprises a TAAG transcription initiation motif flanked by 20 base pairs upstream and 47 base pairs downstream before the native ATG. Many transfer vectors also include a short sequence downstream of the ATG, in which case this sequence is mutated to ATT to abolish translation. However, the ATT sequence, or AUU in the mRNA, is known to be leaky. If a target-coding region is placed in the frame with the AUU, then some products will comprise a chimeric molecule with part of the polyhedrin protein. In this study, we showed that if AUU is placed in the frame with a *Strep* tag and *eGFP* coding region, we could identify a protein product with both sequences present. Further work examined if alternative codons in lieu of AUG might reduce translation initiation further. We found that AUA was used slightly more efficiently than AUU, whereas AUC was the least efficient at initiating translation. The use of this latter codon suggested that there might also be a slight improvement of protein yield if this is incorporated into expression vectors.

**Keywords:** baculovirus expression system; protein expression; protein purification; optimization; bacmid; *flash*BAC; high throughput

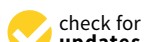

## 1. Introduction

One of the platforms predominantly used for eukaryotic recombinant protein synthesis is the baculovirus expression vector system (BEVS). It utilises recombinant baculoviruses, mainly Autographa californica nucleopolyhedrovirus (AcMNPV) for high-level expression of foreign genes. The BEVS offers several advantages over other expression systems: it is safe to use as baculoviruses exclusively infect arthropods. Large pieces of DNA can be inserted under the control of a variety of strong promoters to control both levels and temporal expression. Very high yields of functional proteins can be achieved, which undergo posttranslational modifications and processing. The manufacturing process is simple, fast, and can be scaled-up easily for insect cells. They can also be used to transduce mammalian cells by expressing the foreign gene under the control of a mammalian-specific promoter. A number of proteins produced by BEVS have been licensed in human and veterinary products, such as vaccines, which emphasises the capability of the system [1,2].

Historically, recombinant baculoviruses were generated by homologous recombination of the circular viral DNA with a transfer vector containing the gene of interest thereby replacing the non-essential polyhedrin (*polh*) gene. However, this was a labour and time intensive process. The process of homologous recombination was very inefficient (<1%).

Several rounds of plaque purification had to be performed to screen for polyhedrin-negative (recombinant) plaques to separate the recombinant from the parental virus [3,4].

This process was improved by the linearisation of the viral DNA prior to co-transfection of cells, which greatly improved the recovery of recombinant viruses to 30% [5]. In a subsequent development, involving the insertion of three *Bsu*36I restriction enzyme sites into the virus genome, the efficiency of recombinant virus production was improved to >90%. However, recombinant viruses still had to be isolated by plaque-purification, albeit aided by a blue/white selection involving removal of *lac*Z from the parental virus genome [6]. The ultimate development of this system was the insertion of a replication-defective baculovirus genome into a low copy number plasmid that could be amplified in *Escherichia coli*. When used to co-transfected cells with a plasmid transfer vector, virus replication was restored with concomitant insertion of the target gene and removal of the bacterial sequences with no need to plaque-purify recombinants [7,8].

In a radically different approach, the need for plaque-purification of recombinant baculoviruses was removed by using a bacterial system to perform the gene insertion step. Maintained in *E. coli*, a bacmid harbours the AcMNPV genome in addition to a mini-F-replicon, kanamycin/chloramphenicol selection markers, a *att*Tn7 site and *lac*Z as a selection marker. All the modifications were inserted into the *polh* locus. Foreign genes can then be inserted into the *att*Tn7 site by site-directed bacterial Tn7 transposition using a donor plasmid that encodes for the gene of interest flanked by Tn7 sites, thereby disrupting the *lac*Z sequence. Cells containing the recombinant bacmid can be screened by blue/white selection. Purified bacmid DNA is then transfected and amplified in insect cells [9].

The *polh* promoter is used in all of the systems to express foreign genes in insect cells. It is active in the very late phase of gene expression in virus-infected cells and naturally directs transcription of the polyhedrin gene. This results in high levels of polyhedrin protein that eventually comprises the major component of occlusion bodies, which serve to transmit viruses between insects in the environment. These structures are not required for virus replication in cell culture; hence, the *polh* coding region can be replaced with foreign sequences. Transcription from the *polh* promoter initiates from a TAAG motif that is conserved in all baculovirus late and very late genes [10]. A 5′ non-coding region of about 46 base pairs is also required for high-level production of recombinant proteins with seven base pairs before the *polh* ATG appearing to have particular significance for high-level gene expression [11,12]. The requirement for sequences upstream of the TAAG motif are less well defined with approximately 20 base pairs required for maximal promoter activity [13].

The 32-base pair region downstream of the ATG codon-encoding methionine is also included in many baculovirus transfer vectors. To avoid translation from the corresponding AUG in the *polh* mRNA, this codon was mutated to ATT [14]. It was noticed that along with the expected recombinant protein, a second slightly larger, but related protein could be detected, when the foreign sequence was cloned in frame with the ATT [15,16]. The undesired protein might co-purify with the recombinant protein of interest and could be challenging to remove in downstream processes. With increasing interest in the use of baculovirus expression vectors for the production of both human and animal vaccines, the issues of product homogeneity and system efficiency becomes very important.

It was suggested that this chimeric protein product might be avoided by inserting the target gene out of frame with the ATT. However, translation could still occur from the AUU codon with the production of a protein of variable length depending on when the first translation stop signal is encountered in the mRNA. Even when the foreign gene is cloned out of frame with ATT, undesired translation from this codon should be avoided to maximise the biosynthesis capacity for the production of the target protein.

Here we report a detailed analysis of the potential for translation from AUU, AUA, and AUC alternatives to AUG in the *polh* leader region. These were compared by the expression of the reporter genes *eGFP* and *lacZ* containing a deletion of their ATG start codon. The least efficient codon for translation initiation occurred from AUC. Subsequently, test expression analyses with target genes with their own ATG were performed comparing

the ATC with the original ATT codon in the *polh* leader region. These experiments indicated that expression levels were higher when a vector with ATC was employed.

## 2. Materials and Methods

### 2.1. Cells, Viruses and Infections

Sf9 cells [17] and *T. ni* High Five^TM (Hi5; BTI-Tn-5B1-4; Invitrogen, Waltham, MA, USA) [18] were maintained in suspension cultures in ESF921 media (Expression Systems, Davis, CA, USA) using standard methods [19]. $1.25 \times 10^5$ Sf9 cells/well and $6.25 \times 10^4$ Hi5 cells/well were seeded in 96-well plates for the detection of eGFP reporter gene expression, while $10^6$ Sf9 cells/well and $0.5 \times 10^6$ Hi5 cells/well were seeded in 12-well plates for the β-galactosidase enzyme assay and protein gel analysis.

### 2.2. Plasmids and Recombinant Viruses

A variant of pOET1 (Oxford Expression Technologies Ltd., Oxford, UK) was constructed with a modified *polyhedrin* gene promoter inserted between the *Eco*RV and *Bam*HI sites comprising 127 base pairs generated from the AcMNPV genomic DNA (coordinates 4428–4554) [1]. This DNA fragment was generated by PCR and utilised a reverse primer that served to mutate the *polyhedrin* ATG codon to ATT. The final plasmid was designated pOET1.ATT. Similar modifications were performed to produce pOET1.ATA, pOET1.ATC, and pOET1.ATG. The multiple cloning site in each of these vectors remained the same as the parental pOET1.

A fragment was excised from pEGFP-N1 (Clontech, Mountain View, CA, USA) using *Bam*HI and *Not*I that contained the *eGFP* coding region. This was inserted into pOET1.ATT to create pOET1.ATT.eGFP. A further modification to this plasmid comprised the insertion of synthetic oligonucleotides at the *Bam*HI site to add a *Strep* tag II coding region [2] that was in frame with the ATT alternative translation initiation codon and the coding region of *eGFP* (pOET1.ATT.*strep*-eGFP). Recombinant viruses (FBU.eGFP and FBU.*Strep*-eGFP) were made using these two plasmids by co-transfecting Sf9 cells with *flash*BAC ULTRA DNA as recommended by the supplier (Oxford Expression Technologies Ltd.).

Using PCR, the 5′ end of the *eGFP* coding region was modified to remove the native ATG sequence and the resulting fragment inserted into pOET1.ATT, pOET1.ATA, pOET1.ATC and pOET1.ATG. This strategy placed the *eGFP* coding region in frame with each of the alternative translation initiation codons. Each plasmid was then used to produce the recombinant viruses FBU.ATTΔATG.eGFP, FBU.ATAΔATG.eGFP, FBU.ATCΔATG.eGFP, and FBU.ATGΔATG.eGFP using *flash*BAC ULTRA as before. A similar modification was done with the *lacZ* coding region from pCH110 [3] to produce FBU.ATTΔATG.lacZ, FBU.ATAΔATG.lacZ, FBU.ATCΔATG.lacZ, and FBU.ATGΔATG.lacZ.

The *eGFP* and *lacZ* coding regions were inserted into pOET1.ATT and pOET1.ATC so that they were out of frame with the alternative translation initiation codon. These plasmids were used to produce the recombinant viruses FBU.ATT.eGFP, FBU.ATC.eGFP, FBU.ATT.lacZ, and FBU.ATC.lacZ with *flash*BAC ULTRA DNA. A synthetic copy of the Crimean–Congo haemorrhagic fever virus nucleoprotein gene [4] with a six-histidine tag at its 5′ end was constructed and inserted into pOET1.ATT and pOET1.ATC and used to make the viruses FBU.ATT.CCHF-NP and FBU.ATC.CCHF-NP.

Viruses were produced by co-transfecting Sf9 cells with *flash*BAC ULTRA (FBU) and various modified transfer vectors based upon pOET1 (Oxford Expression Technologies Ltd.), followed by virus amplification [19]. Virus titres were determined by qPCR as described previously [20]. Cells were infected with the indicated recombinant virus at a multiplicity of infection (MOI) of 5 [19].

### 2.3. Protein Synthesis Analysis

#### 2.3.1. Detection of eGFP Fluorescence

Relative eGFP fluorescence intensity ($\lambda_{Emission}$ = 485 nm, $\lambda_{Extinction}$ = 535 nm) was measured in a microplate Spectrophotometer (SpectraMax i3, Molecular Devices, San Jose,

CA, USA) at the indicated time points. Background relative fluorescent units (RFU; from mock-infected samples) was subtracted. Experiments were performed three times with each containing four replicates.

### 2.3.2. Detection of β-Galactosidase

Cells were harvested at the indicated time points by centrifugation at 4000 g for 2 min, after which they were lysed by performing three freeze/thaw cycles. The β-galactosidase assay was performed as described previously [21] in 96-well plates. Experiments were performed twice in duplicate. Absorbance of each sample was then analysed in triplicate at $OD_{405nm}$ in a microplate absorbance reader (LT-4500, Labtech, Heathfield, UK). Background absorbance (from mock-infected samples) was subtracted.

### 2.3.3. SDS-PAGE, Coomassie Blue Staining, Western Blotting

Cells were harvested at indicated time points by centrifugation at $4000\times g$ for 2 min. Subsequently, cell pellets were resuspended in Milli-Q $H_20$ and boiled for at least 5 min in $5 \times$ SDS loading buffer (50 mM Tris-HCl pH 6.8, 2% (*w/v*) SDS, 10% (*v/v*) glycerol, 0.02% (*w/v*) bromophenol blue, 5% (*v/v*) β-mercaptoethanol). Proteins were visualised either by Coomassie Brilliant Blue staining or by immunoblotting against the protein of interest using standard methods [21]. PVDF membranes were probed with either anti GFP rabbit polyclonal (Abcam), anti β-galactosidase rabbit polyclonal (Life Technologies, Carlsbad, CA, USA), anti-Histidine-tagged mouse monoclonal (Bio-Rad AbD Serotec, Raleigh, NC, USA), or anti Strep-tag classic mouse monoclonal (Bio-Rad) antibodies. Goat anti mouse or rabbit IgG Alkaline Phosphatase were purchased from Sigma-Aldrich. Images were acquired by ChemiDoc imaging systems (Bio-Rad AbD Serotech) and processed with Image Lab software (version 6.1, Bio-Rad AbD Serotec). Experiments were performed twice in duplicate.

### *2.4. Statistics*

Statistical analysis was performed using GraphPad Prism (version 9.2.0).

## **3. Results**

### *3.1. Protein Synthesis Is Initiated at AUU in Polh 5′ UTR*

Initially, we determined if translation could initiate from the AUU codon in the *polh* mRNA produced by many baculovirus expression vectors. For this, the *eGFP* coding sequence was cloned in frame with the ATT codon in the 5′ UTR of *polh*. To visualise the expression of both native protein and the predicted polh-eGFP fusion protein, the *polh* leader was further modified to insert a *Strep*-tag II coding region in frame between the ATT and the ATG of *eGFP* (Figure 1a). Sf9 and Hi5 cells were infected with the recombinant virus FBU.ATT.*Strep*.eGFP and cell lysates analysed by immunoblotting against the EGFP and Strep tag at 72 h post infection (hp.i.) (Figure 1b,c). While a large amount of eGFP accumulated, a smaller amount of the *Strep*-tagged polh-eGFP could be observed in both insect cell lines analysed, although this result is not quantitative. As expected, no *Strep*-tagged protein could be detected after mock, FBU.Null or FBU.eGFP virus infections.

### *3.2. Protein Translation Is Initiated at AUU, AUC, or AUA in the polh 5′ UTR*

After confirming that a small amount of protein was translated from AUU present in the *polh* leader region, it was investigated if initiation occurs when the AUU codon was modified to either AUC or AUA. Finding the weakest initiator codon would help in redesigning expression vectors to minimise the premature start of translation. To test this hypothesis, the reporter gene *eGFP* lacking the ATG start codon was cloned in frame with ATT, ATC, ATA or as a positive control with an ATG codon in the *polh* 5′UTR (Figure 2a). Hence, *eGFP* expression would only be detectable if initiated from the leader region. Sf9 and Hi5 cells were infected with the recombinant viruses FBU.ATGΔATG.eGFP, FBU.ATTΔATG.eGFP, FBU.ATCΔATG.eGFP, and FBU.ATAΔATG.eGFP and eGFP synthesis was measured as

RFU between 24–96 hp.i. (Figure 2b,c). High levels of eGFP fluorescence were measured after infection with the positive control virus FBU.ATGΔATG.eGFP. When the *polh* initiation codon was modified to any of the other three codons mentioned above, eGFP fluorescence was above levels of that of the negative control (FBU.Null infection) from 48 hp.i., but was about 10-fold lower than the positive control (Table 1). It was further noticed that the least amount of eGFP fluorescence was detectable when the initiation codon in the 5′UTR was modified to ATC in both cell lines analysed.

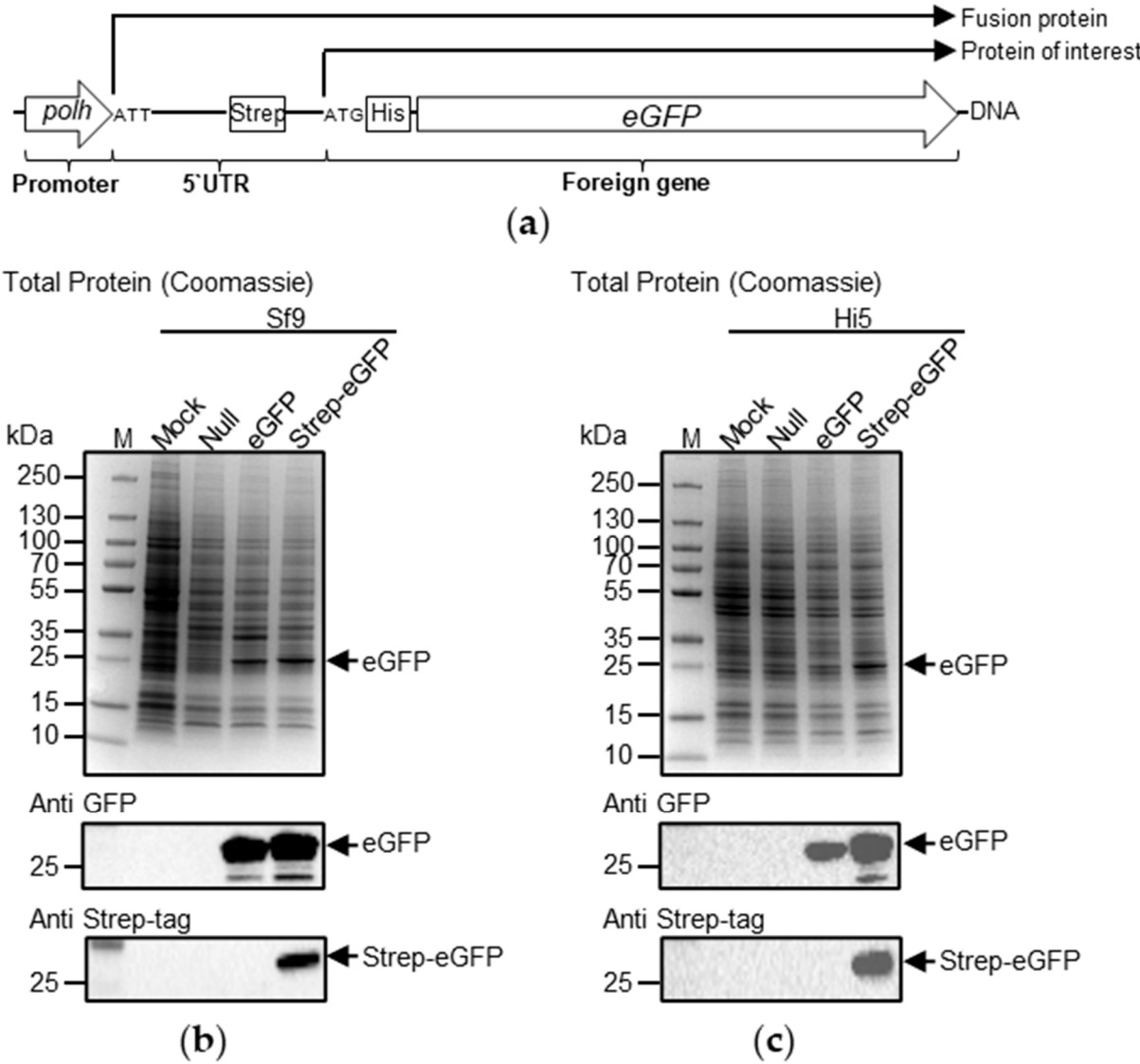

**Figure 1.** Protein translation initiation from the AUU codon in the *polh* 5′UTR. (**a**) The coding region of the reporter gene eGFP was cloned in frame with the ATT codon in the 5′UTR of *polh*. The *polh* leader region was further modified by introducing a *Strep*-tag II in frame between ATT and ATG of *eGFP*. (**b**) Sf9 or (**c**) Hi5 cells were infected with FBU.Null, FBU.eGFP or FBU.ATT.*Strep*.eGFP at MOI 5 or mock-infected. Cell lysates were analysed by SDS-PAGE followed by a Coomassie stain (upper panel) or immunoblotting against eGFP or Strep-tag II (lower panel).

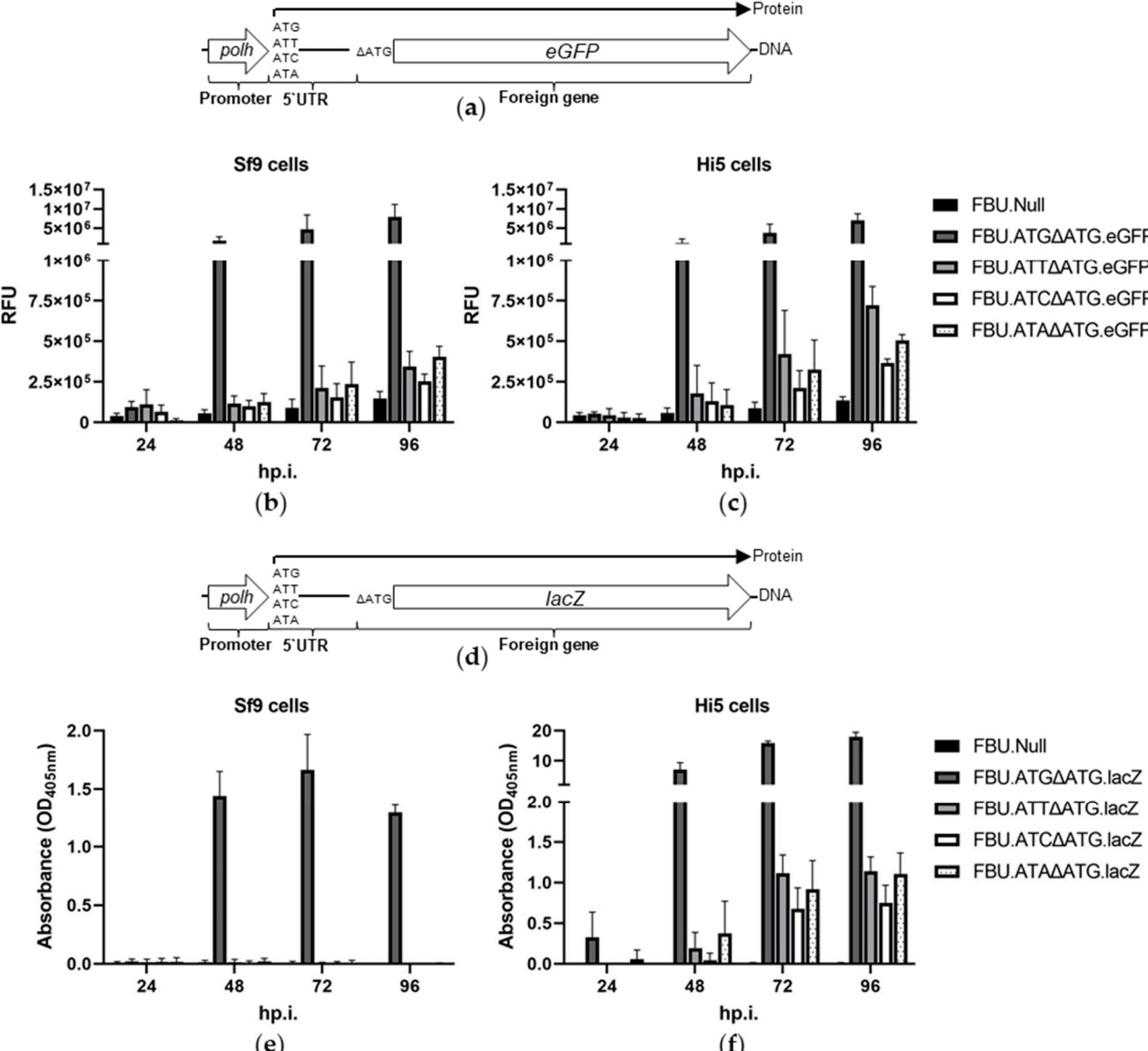

**Figure 2.** Translation of mRNA from alternative initiation codons. The coding region of the reporter genes (**a**) *eGFP* or (**d**) *lacZ* containing a deletion of the ATG start codon was cloned in frame with each of the following codons in the *polh* leader region: ATG, ATT, ATC or ATA. (**b**) Sf9 or (**c**) Hi5 cells were infected with FBU.Null, FBU.ATGΔATG.eGFP, FBU.ATTΔATG.eGFP, FBU.ATCΔATG.eGFP and FBU.ATAΔATG.eGFP at MOI 5 or mock-infected. *EGFP* expression was measured as relative fluorescent units at indicated time points, p.i., by a microplate spectrophotometer. Background readings were subtracted from all results. Alternatively, (**e**) Sf9 or (**f**) Hi5 cells were infected with FBU.Null, FBU.ATGΔATG.lacZ, FBU.ATTΔATG.lacZ, FBU.ATCΔATG.lacZ, and FBU.ATAΔATG.lacZ at MOI 5 or mock-infected. *LacZ* expression was measured by a β-galactosidase enzyme assay using an absorbance microplate reader at OD405 nm. Background was subtracted. Bars represent average between three independent experiments carried out in quadruplet. Error bars show the standard deviation.

**Table 1.** Translation of mRNA from alternative initiation codons [1].

| | Sf9 eGFP | | | |
|---|---|---|---|---|
| | FBU.ATG ΔATG.eGFP | FBU.ATT ΔATG.eGFP | FBU.ATC ΔATG.eGFP | FBU.ATA ΔATG.eGFP. |
| 24 hp.i | 100% | 117.2% | 70.1% | 13.8% |
| 48 hp.i. | 100% | 6.2% | 5.4% | 6.8% |
| 72 hp.i. | 100% | 4.5% | 3.3% | 5.0% |
| 96 hp.i. | 100% | 4.3% | 3.2% | 5.1% |
| | Hi5 eGFP | | | |
| | FBU.ATG ΔATG.eGFP | FBU.ATT ΔATG.eGFP | FBU.ATC ΔATG.eGFP | FBU.ATA ΔATG.eGFP |
| 24 hp.i | 100% | 80.2% | 56.5% | 49.0% |
| 48 hp.i. | 100% | 15.3% | 11.1% | 9.1% |
| 72 hp.i. | 100% | 11.2% | 5.6% | 8.5% |
| 96 hp.i. | 100% | 10.3% | 5.2% | 7.2% |
| | Sf9 β-Galactosidase | | | |
| | FBU.ATG ΔATG.lacZ | FBU.ATT ΔATG.lacZ | FBU.ATC ΔATG.lacZ | FBU.ATA ΔATG.lacZ |
| 24 hp.i | 100% | 69.6% | 77.1% | 88.7% |
| 48 hp.i. | 100% | 1.0% | 0.7% | 1.4% |
| 72 hp.i. | 100% | 0.3% | 0.4% | 0.7% |
| 96 hp.i. | 100% | 0.1% | 0.0% | 0.2% |
| | Hi5 β-Galactosidase | | | |
| | FBU.ATG ΔATG.lacZ | FBU.ATT ΔATG.lacZ | FBU.ATC ΔATG.lacZ | FBU.ATA ΔATG.lacZ |
| 24 hp.i | 100% | 0.0% | 0.0% | 17.2% |
| 48 hp.i. | 100% | 2.8% | 0.6% | 5.3% |
| 72 hp.i. | 100% | 7.1% | 4.3% | 5.8% |
| 96 hp.i. | 100% | 6.4% | 4.2% | 6.2% |

[1] Percentage of eGFP and β-galactosidase expression in virus-infected Sf9 or Hi5 cells are compared to the positive control.

To confirm these results, similar recombinant viruses were generated, but this time expression of the reporter gene *lacZ* was monitored (Figure 2d). Again, the *lacZ* start codon ATG was deleted and the *polh* leader region was modified containing the initiation codons ATG, ATT, ATC or ATA. Insect cells were infected with the recombinant viruses FBU.ATGΔATG.lacZ, FBU.ATTΔATG.lacZ, FBU.ATCΔATG.lacZ, and FBU.ATAΔATG.lacZ and *lacZ* expression was analysed by a β-galactosidase enzyme assay measuring absorbance at $OD_{405nm}$ between 24–96 hp.i. (Figure 2e,f). High levels of β-galactosidase enzyme activity were present after infection with the positive control virus containing ATG in the *polh* 5′UTR. When this codon was modified to ATT, ATC, or ATA, no enzyme activity above negative control levels (FBU.Null infections) could be detected at any time point analysed p.i. in virus-infected Sf9 cells. However, absorbance levels were above that of the negative control in virus-infected Hi5 cells. Similar to results described above, the least amount of β-galactosidase enzyme activity could be detected after the initiation codon was changed to ATC (Table 1). Although not significant, this reduced translation from the AUC codon was consistent in repeated experiments.

### 3.3. The Effect of Substituting AUC for AUU on Recombinant Protein Production

While replacing the ATT codon in the *polh* 5′ UTR reduced premature translation, we were concerned that it might also adversely affect recombinant protein yield. This region is known to be very important for *polh* gene expression [11,15,22]. To analyse this, recombinant viruses were generated expressing the reporter gene *eGFP* with its own ATG

under the control of the *polh* promoter. The *polh* leader region either contained the initiation codon ATT as present in commercially available BEVS vectors or was modified to ATC (Figure 3a). Neither ATT nor ATC were in frame with *eGFP*. Insect cells were infected with FBU.ATT.eGFP, and FBU.ATC.eGFP and eGFP expression was analysed as RFU between 24–96 hp.i. (Figure 3b,c). There was a tendency for higher eGFP fluorescence at all time points analysed from insect cells infected with viruses containing the ATC codon in the *polh* 5′ UTR rather than the ATT, but the differences were not statistically significant. The same trend was observed when the amount of eGFP was determined by protein gels followed by a Coomassie stain or immunoblotting against eGFP (Figure 3d,e).

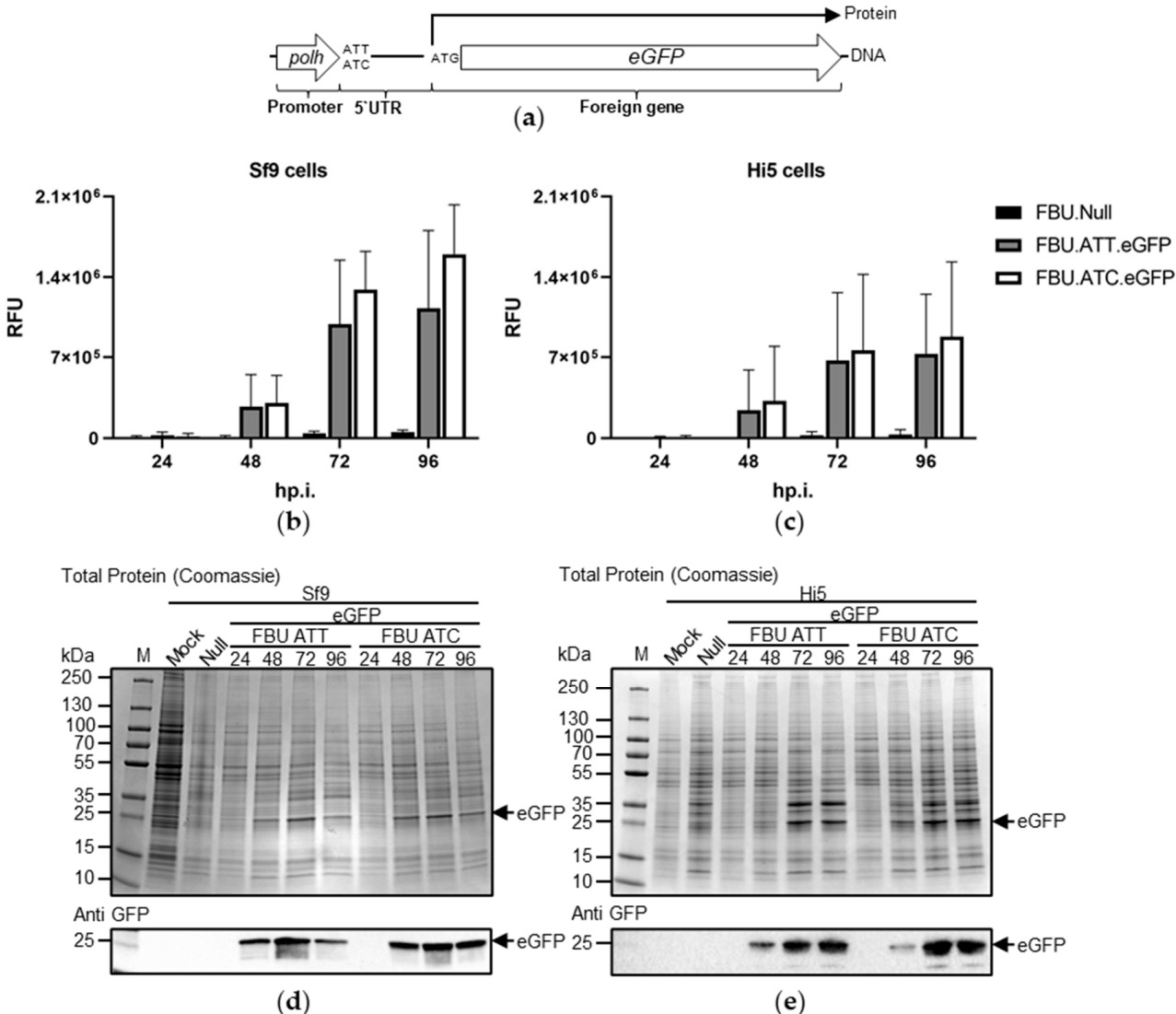

**Figure 3.** EGFP expression analysis from recombinant baculoviruses containing a modified *polh* 5′ UTR. (**a**) The coding region of the reporter gene *eGFP* was cloned under the control of the *polh* promoter. The *polh* leader region contained either the initiation codon ATT or a modification of the codon to ATC. (**b,d**) Sf9 and (**c,e**) Hi5 cells were infected with FBU.Null, FBU.ATT.eGFP and FBU.ATC.eGFP at MOI 5 or mock-infected. (**b,c**) eGFP expression was measured as relative fluorescent units at indicated time points, p.i. by a microplate spectrophotometer. Background (mock-infection) was subtracted. Bars represent the average of three independent experiments carried out in quadruplet. Error bars show standard deviations. (**d,e**) Alternatively, virus-infected cell lysates were analysed by SDS-PAGE followed by a Coomassie stain (upper panel) or immunoblotting against eGFP (lower panel). Representative images of protein gels carried out as two independent experiments in duplicate are shown.

To confirm these results, recombinant viruses expressing the reporter gene *lacZ* were utilised in similar experiments (Figure 4a). β-galactosidase enzyme activity was measured in FBU.ATT.lacZ and FBU.ATC.lacZ-infected insect cells between 24–96 hp.i. in a microplate absorbance reader (Figure 4b,c). Results showed slightly higher enzyme activity after FBU.ATC.lacZ than FBU.ATT.lacZ-infections in Sf9 cells. However, this was not mirrored in virus-infected Hi5 cells. Here, the ATT codon resulted in slightly higher *lacZ* expression up to 72 hp.i. Higher levels of β-galactosidase were then again measured at 96 hp.i. after FBU.ATC.lacZ-infections. Once again, none of these results were statistically significant. When *lacZ* expression was analysed by SDS-PAGEs with a subsequent Coomassie stain or immunoblotting against β-galactosidase, higher accumulations of β-galactosidase were observed when the *polh* leader region was modified to contain the ATC than the ATT codon in both Sf9 and Hi5-infected cell lysates (Figure 4d,e).

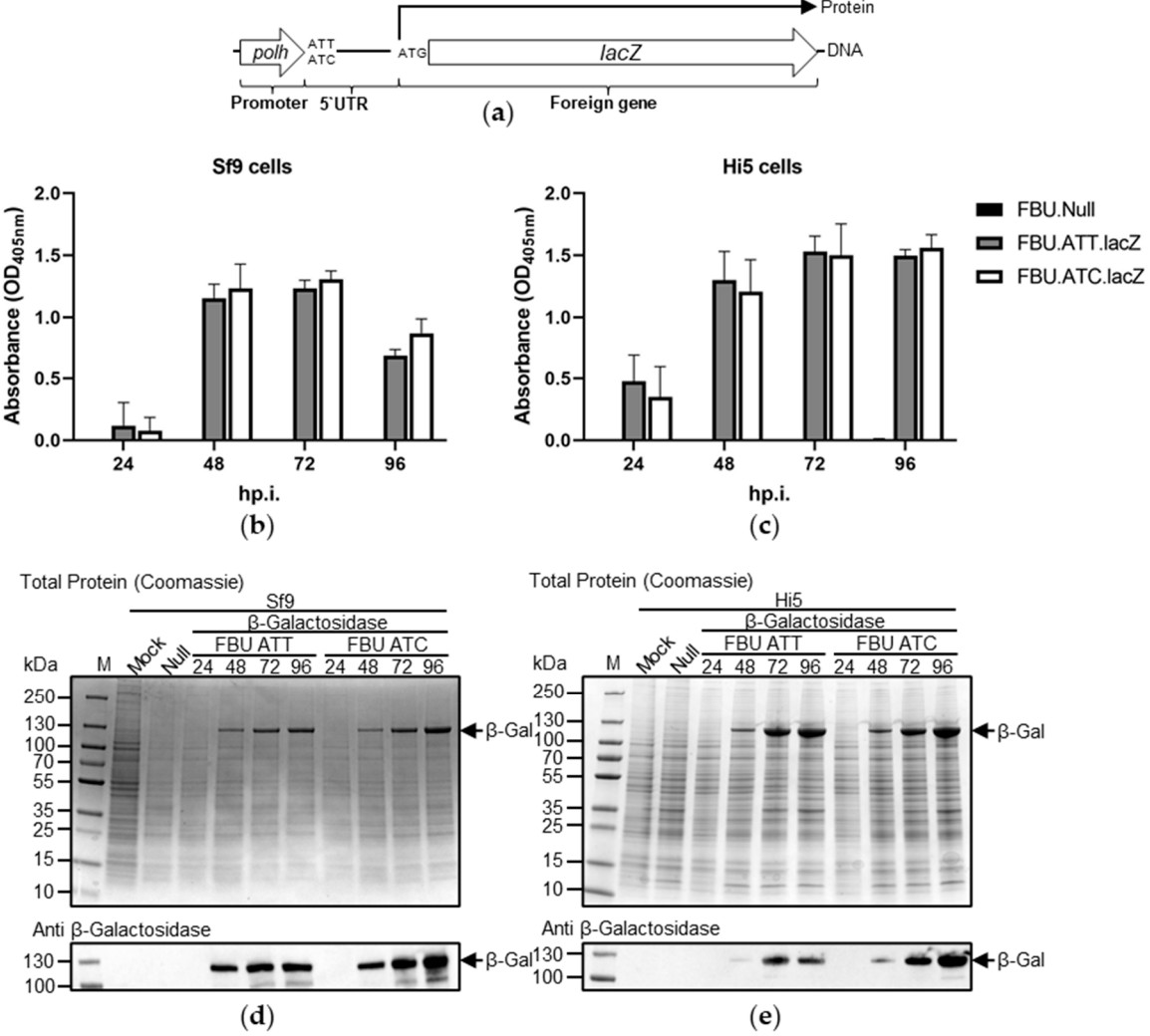

**Figure 4.** β-Galactosidase expression analysis from recombinant baculoviruses containing a modified *polh* 5′ UTR. (**a**) The coding region of the reporter gene *lacZ* was cloned under the control of the *polh* promoter. The *polh* leader region contained either the initiation ATT codon or a modification of the codon to ATC. (**b,d**) Sf9 and (**c,e**) Hi5 cells were infected with FBU.Null, FBU.ATT.lacZ and FBU.ATC.lacZ at MOI 5 or mock-infected. (**b,c**) *LacZ* expression was measured by a β-galactosidase enzyme assay using an absorbance microplate reader at OD405 nm. Background (mock-infection) was subtracted. Bars represent the average of two independent experiments performed in duplicate. Each of the samples was then assayed in triplicate using the β-galactosidase enzyme assay. (**d,e**) Alternatively, virus-infected cell lysates were analysed by SDS-PAGE followed by a Coomassie stain (upper panel) or immunoblotting against β-Galactosidase (lower panel). Representative images of protein gels carried out as two independent experiments in duplicate are shown.

Thus far, the effect of ATT and ATC in the *polh* 5′UTR on gene expression were determined by analysing reporter gene expression. However, reporter genes are typically well expressed. Therefore, we investigated the effect of these two codons on the target gene Crimean–Congo haemorrhagic fever virus (CCHFv) nucleoprotein (NP) tagged with six histidine amino acids at the amino terminus (Figure 5a). Insect cells were infected with recombinant viruses expressing CCHFv NP, in which the *polh* leader region contained either the ATT codon or this codon modified to ATC. Protein synthesis was analysed by SDS-PAGE, followed by either Coomassie staining or immunoblotting against the NP his-tag (Figure 5b,c). There was no detrimental effect on NP production by utilization of ATC instead of ATT in the 5′ *polh* UTR.

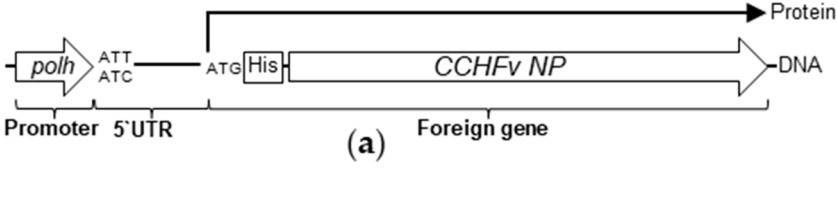

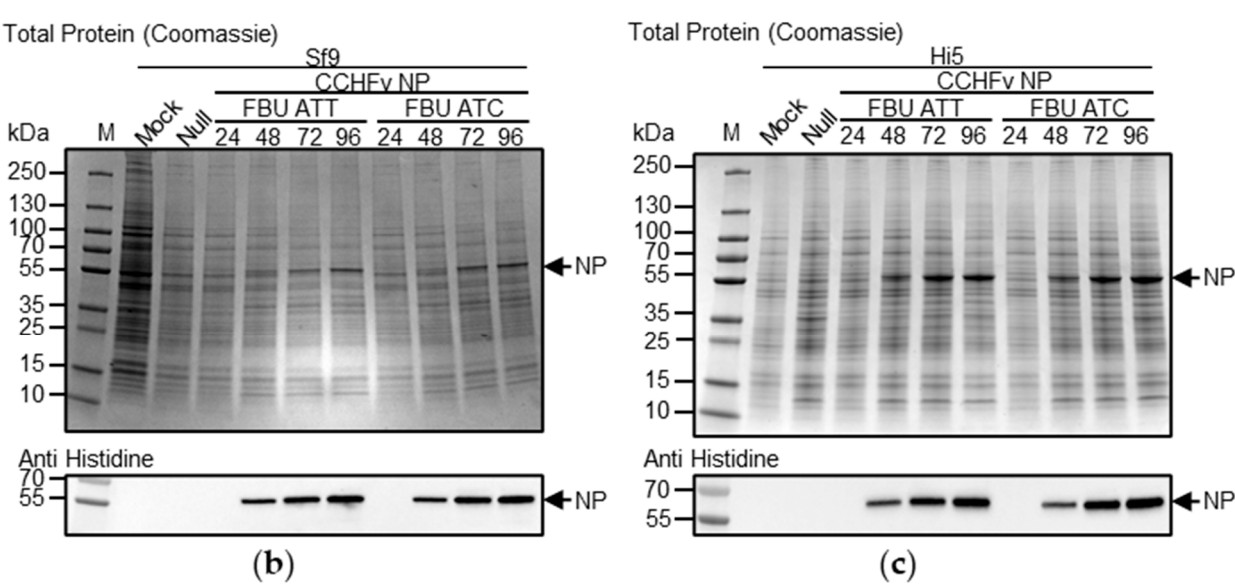

**Figure 5.** CCHFv NP expression analysis from recombinant baculoviruses containing a modified *polh* 5′UTR. (**a**) The coding region of the reporter gene *CCHF NP* was cloned under the control of the *polh* promoter. The *polh* leader region contained either the original initiation ATT codon or a modification of the codon to ATC. (**b**) Sf9 or (**c**) Hi5 cells were infected with FBU.Null, FBU.ATT.CCHF_NP and FBU.ATC. CCHF_NP at MOI 5 or mock-infected. (**b,c**) Virus-infected cell lysates were analysed by SDS-PAGE followed by a Coomassie stain (upper panel) or immunoblotting using an antibody to 6-his (lower panel). Representative images of protein gels carried out as two independent experiments performed in duplicate are shown.

## 4. Discussion

The BEVS has been used widely and is extremely successfully as an eukaryotic expression system for the production of recombinant proteins for nearly 40 years. Medical and therapeutic applications of these proteins have been developed more slowly, but there are now two vaccines (Cervarix® for HPV and FluBlok® for influenza virus) on the market with undoubtedly more products under development. Although highly improved during the last few decades from the point of view of ease of use for non-specialists, challenges in the application of the system remain. One obstacle to solve is the undesired translation from the *polh* initiator codon that was altered to AUU in commercially available BEVS [15]. This can lead to undesired fusion proteins and might impact the purity of the target protein (Figure 1) [16].

The production of proteins as a result of errors in translation initiation can lead to molecules that aggregate, misfold, or simply contain extra amino acids that compromise quality [23]. Such errors in translation initiation were first identified in the 1980s when it was discovered that codons other than AUG could be used in mRNA to start protein synthesis [24–28]. The context of an initiation codon is also important. A functional role was identified for the nucleotide -3 relative to the AUG initiation codon [29]. Further studies defined the "Kozak sequence" as C(A/G)CCAUGG in mammals [30]. In AcMNPV, an analysis of nucleotide frequencies for start codons of 154 selected open reading frames suggested a consensus of AA(A/C)AUGA [31].

Ribosome profiling has also demonstrated that thousands of novel initiations may occur at non-AUG codons [32,33]. Various other studies have shown that not all alternative initiation codons are used with equal efficiency in different systems. Overall, CUG is used the most frequently, followed by GUG, ACG, and AUU [34].

Translation initiation from AUU in viruses is not a phenomenon exclusively observed in baculoviruses but has also been recognised in other viruses, such as Rice Tungro Bacilliform Virus (RTBV) [35]. Other potential non-AUG initiation codons have also been identified, e.g., the RNA2 of THE soil-borne wheat mosaic virus (SBWMV) encodes for a protein initiated at CUG [36].

In our study, we used the *eGFP* coding region fused in frame with an ATT codon and *Strep* tag to demonstrate that translation could initiate from this position, albeit at an apparently low level. A quantitative analysis of the phenomenon was then conducted by fusing ATT in frame with *eGFP* or *lacZ* that lacked their native ATG codons. We also included the potential alternative codons AUC and AUA as well as AUG in this experiment. The results indicated that the least translation occurred from AUC in the *polh* 5′UTR (Figure 2). Translation from each of the three potential alternative codons was initiated but not as efficiently as from the conventional start codon AUG. In one of our downstream assays, we could not detect any *lacZ* gene expression above background in virus-infected Sf9 with any of the three modified initiation codons (Figure 2d). However, the β-galactosidase assay might not have been sensitive enough to detect such small variations in enzyme activity.

Generally, initiation from codons other than AUG is rare in eukaryotes [37]. Studies in prokaryotes suggested that AUG forms the most stably interaction with the CAU anticodon in t-RNAs and is therefore the most common initiator sequence [38]. However, eukaryotic ribosomes might be able to initiate at non-AUG codons depending on the sequence context and thereby compensating for the weak codon-anticodon interaction [30]. Overall, our results re-emphasise the importance of cloning any target gene out of frame with the modified *polh* initiator codon.

As changing the *polh* initiator codon to ATC did not abolished translation completely, we modified the leader region and introduced a stop codon in the 5′UTR in frame with the ATT codon. Surprisingly, expression of reporter genes from these recombinant viruses was reduced when compared to viruses not encoding for the stop codon in the *polh* 5′UTR (data not shown). However, not only stability of the AUG codon-anticodon stability determines high expressions of proteins, but also the immediate sequence context. Partial deletions of the *polh* leader as well as substituted DNA adversely affected levels of mRNA and therefore the yield of protein production [15].

Given the importance of the 5′UTR for expression using the *polh* promoter, it was important to determine that utilizing ATC instead of ATT did not have an adverse effect on recombinant protein production. Modifications of the *polh* leader from ATT to ATC resulted in similar recombinant protein expression levels. Our results pointed to a small benefit of having ATC over ATT on the amount of expression of eGFP, β-Galactosidase and CCHF NP (Figures 3–5). While the increases in yield for each protein were only modest, in the context of commercial production even an increase of a few percentage points can be economically significant in the longer term.

While using ATC rather than ATT in the 5′ *polh* UTR did still result in some translation initiation, there is scope to reduce this further. Many combinations of alternative ATG codons could be tested empirically in the baculovirus-insect cell expression system given that, in that other species, wide variation is seen in the efficiency with which such codon variants can be utilised.

In summary, this work has specified new strategies for designing improved transfer vectors for the optimal production of recombinant proteins using the BEVS.

**Author Contributions:** Conceptualization, R.D.P., L.A.K., A.C.C., C.B., L.P.G., S.L. and D.R.B.; methodology, C.B., D.R.B., G.B., S.L. and E.G.; formal analysis, C.B., D.R.B., A.C.C., L.P.G., R.D.P. and L.A.K.; data curation, C.B. and D.R.B.; writing—original draft preparation, C.B. and R.D.P.; writing—review and editing, L.A.K.; visualization, C.B.; supervision, R.D.P., A.C.C., L.P.G., C.B., R.A.-G. and L.A.K.; project administration, R.D.P. and L.A.K.; funding acquisition, S.L. All authors have read and agreed to the published version of the manuscript.

**Funding:** L.S. was funded through the China Scholarship Council (funding number 2018/08440611).

**Institutional Review Board Statement:** Not applicable.

**Informed Consent Statement:** Not applicable.

**Data Availability Statement:** The data underpinning what was presented in this study are available upon request from the corresponding author.

**Conflicts of Interest:** The authors declare no conflict of interest. The funder had no role in the design of the study; in the collection, analyses, or interpretation of data; in the writing of the manuscript, or in the decision to publish the results.

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
