# Peer review of "Optimizing Recombinant Baculovirus Vector Design for Protein Production in Insect Cells"

_processes, doi:10.3390/pr9122118_

Round 1

Reviewer 1 Report

The manuscript describes an approach for re-designing baculovirus expression vectors. The most frequently used viral vectors are based on the very late polyhedrin gene promoter, many of which include parts of the polyhedrin coding sequence downstream the native translation start codon. To avoid unwanted fusion proteins between the target gene and polyhedrin residues, the start signal is replaced by ATT. Despite this mutation some fusion protein can arise when the gene of interest is placed in frame with the ATT codon. Using a model protein, the authors showed that such a sequence arrangement leads to both types of protein, even if the fusion construct is present at small amounts. In further experiments they compared three alternative codons against the standard initiation codon ATG and found out that ATC was the least efficient at initiating translation, suggesting its potential use for expression vector engineering.

The report at hand is very well written and easy to follow. The scientific problem is narrowly specified and can be investigated with easy to perform reporter assays. The effect of alternative start codons on reporter gene expression is examined by enzyme activity and protein mass expressed. Correct size and protein quality are detected by SDS-PAGE and Western blotting. These experimental procedures are carefully designed and performed with scientific soundness. Nevertheless, it is suggested to include mRNA values of the promoter constructs in the study to provide details regarding potential deviations between protein and mRNA expression levels.

One outcome of the study is: the ATT codon, frequently used in transfer vectors, can serve as translation start signal when placed in frame with the gene of interest. This has been known before and here the authors provide immune blots of the fusion product.

The central question of this study is: can replacement of ATT by alternative codons decrease early translation? All tested sequences resulted in similar expression levels with ATC being the least efficient. Statistical analysis was performed with an adequate number of biological and technical replicates. However, the effect of codon ATC in suppressing premature translation was not significant. In the discussion this aspect is sufficiently addressed. In this respect, re- engineering of transfer vectors is questionable when improvement of protein yield is anticipated.

Eventually, the manuscript can be recommended for publication.

minor issues:

 In legend figure 2,

line 225: “FBU.ATGdeltaATC.eGFP” should read “FBU.ATCdeltaATG.eGFP”

line 228: same for construct “FBU.ATGdeltaATC.lacZ”

line 314: change “Figure 4” to “Figure 5”

line 360: change “Figure 4e” to “figure 2d”

Author Response

Optimizing recombinant baculovirus vector design for protein production in insect cells

Manuscript ID: processes-1462725

Comments to Reviewer 1:

We appreciate the time and effort that you have dedicated in providing your feedback on our manuscript. We are very grateful for your comments and suggestions. Please find below our response to your comments.

Nevertheless, it is suggested to include mRNA values of the promoter constructs in the study to provide details regarding potential deviations between protein and mRNA expression levels.

Many thanks for your suggestion. It would have been interesting to explore this aspect. However, in case of our study, we focused on the effect of promoter modifications on the level of proteins for potential benefits in commercial production. Even small increases in the quality and/or yield of a recombinant proteins may account for reduced downstream processing and may be economically more sustainable. Nevertheless, we will keep your comment in mind for future research.

In addition all minor issues pointed out by the reviewer have been addressed and tracked changed.

Reviewer 2 Report

still 2 questions should be addressed:

  1. The house keeping gene should be performed as internal reference in each experiment to normalize the different samples, so that we can compare the target genes expression in different samples.
  2. What about the results if the ATG downstream of the target gene was mutated?

Author Response

Optimizing recombinant baculovirus vector design for protein production in insect cells

Manuscript ID: processes-1462725

Comments to Reviewer 2:

The authors would like to thank the Reviewer for his/her comments and feedback on our manuscript. Please find our responses to your comments below. All changes made to the manuscript have been tracked.

The house keeping gene should be performed as internal reference in each experiment to normalize the different samples, so that we can compare the target genes expression in different samples.

You have raised an important point here. However, we believe that the stained total protein gels will give insights into loading equality and hence target genes expression between different samples. The use of housekeeping genes, or in this case housekeeping proteins, is problematic very late in virus infection as host gene expression is vastly downregulated at these times. 

What about the results if the ATG downstream of the target gene was mutated?

This is an interesting question. However, while there are examples in the literature, in which protein translation was initiated at the second ATG, this question might be slightly out of scope for our study. Mutating the ATG downstream of the target gene will change the protein composition and might change its structure and/or characteristics.  ATG (AUG) only specifies methionine so you can’t perform a silent mutation as with other amino acids. 

Round 2

Reviewer 2 Report

No comments